# MINIBATCH STOCHASTIC THREE POINTS METHOD FOR UNCONSTRAINED SMOOTH MINIMIZATION

## ABSTRACT

In this paper, we propose a new zero order optimization method called *minibatch stochastic three points (`MiSTP`)* method to solve an unconstrained minimization problem in a setting where only an approximation of the objective function evaluation is possible. It is based on the recently proposed stochastic three points (`STP`) method (Bergou et al., 2020). At each iteration, `MiSTP` generates a random search direction in a similar manner to `STP`, but chooses the next iterate based solely on the approximation of the objective function rather than its exact evaluations. We also analyze our method's complexity in the nonconvex and convex cases and evaluate its performance on multiple machine learning tasks.

## 1 INTRODUCTION

In this paper we consider the following unconstrained finite-sum optimization problem:

$$\min_{x \in \mathbb{R}^d} f(x) \stackrel{\text{def}}{=} \frac{1}{n} \sum_{i=1}^{n} f_i(x) \tag{1}$$

where each $f_i : \mathbb{R}^d \to \mathbb{R}$ is a smooth objective function. Such kind of problems arise in a large body of machine learning (ML) applications including logistic regression (Conroy & Sajda, 2012), ridge regression (Shen et al., 2013), least squares problems (Suykens & Vandewalle, 1999), and deep neural networks training. The formulation (1) can express the distributed optimization problem across $n$ agents, where each function $f_i$ represents the objective function of agent $i$, or the optimization problem where each $f_i$ is the objective function associated with the data point $i$. We assume that we work in the Zero Order (ZO) optimization settings, i.e., we do not have access to the derivatives of any function $f_i$ and only functions evaluations are available. Such situation arises in many fields and may occur due to multiple reasons, for example: (i) In many optimization problems, there is only availability of the objective function as the output of a black-box or simulation oracle and hence the absence of derivative information (Conn et al., 2009). (ii) There are situations where the objective function evaluation is done through an old software. Modification of this software to provide first-order derivatives may be too costly or impossible (Conn et al., 2009; Nesterov & Spokoiny, 2017). (iii) In some situations, derivatives of the objective function are not available but can be extracted. This necessitates access and a good understanding of the simulation code. This process is considered invasive to the simulation code and also very costly in terms of coding efforts (Kramer et al., 2011). (IV) In the case of using a commercial software that evaluates only the functions, it is impossible to compute the derivatives because the simulation code is inaccessible (Kramer et al., 2011; Conn et al., 2009). (V) In the case of having access only to noisy function evaluations, computing derivatives is useless because they are unreliable (Conn et al., 2009). ZO optimization has been used in many ML applications, for instance: hyperparameters tuning of ML models (Turner et al., 2021; P.Koch et al., 2018), multi-agent target tracking (Al-Abri et al., 2021), policy optimization in reinforcement learning algorithms (Malik et al., 2020; Li et al., 2020), maximization of the area under the curve (AUC) (Ghanbari & Scheinberg, 2017), automatic speech recognition (Watanabe & Roux, 2014), and the generation of black-box adversarial attacks on deep neural network classifiers (Ughi et al., 2021). Google Vizier system (Golovin et al., 2017) which is the de facto parameter tuning engine at Google is also based on ZO optimization.

There exist many ZO methods that solve problem (1), most of them approximate the gradient using gradient smoothing techniques such as the popular two-point gradient estimator (Nesterov &

Spokoiny, 2017). Ghadimi & Lan (2013) proposed a stochastic version of the algorithm proposed by Nesterov & Spokoiny (2017) (called `RSGF`) in the case of function values being stochastic rather than deterministic. Liu et al. (2018) also proposed a ZO stochastic variance reduced method (called `ZO-SVRG`) based on the minibatch variant of `SVRG` method (Reddi et al., 2016). `ZO-SVRG` can use different gradient estimators namely RandGradEst, Avg-RandGradEst, and CoordGradEst presented in Liu et al. (2018). Another popular class of ZO methods is Direct-Search (`DS`) methods. They determine the next iterate based solely on function values and does not develop an approximation of the derivatives or build a surrogate model of the the objective function (Conn et al., 2009). For a comprehensive view about classes of ZO methods we refer the reader to a survey by Larson et al. (2019). More related to our work, Bergou et al. (2020) proposed a ZO method called Stochastic Three Points (`STP`) which is a general variant of direct search methods. At each training iteration, `STP` generates a random search direction $s$ according to a certain probability distribution and updates the iterate as follow:

$$x = \arg\min\{f(x - \alpha s), f(x + \alpha s), f(x)\}$$

where $\alpha > 0$ is the stepsize. `STP` is simple, very easy to implement, and has better complexity bounds than deterministic direct search (`DDS`) methods. Due to its efficiency and simplicity, `STP` paved the way for other interesting works that are conducted for the first time, namely the first work on importance sampling in the random direct search setting ( $STP_{IS}$ method) (Bibi et al., 2020) and the first ZO method with heavy ball momentum (`SMTP`) and with importance sampling ($SMTP_{IS}$) (Gorbunov et al., 2020). To solve problem (1), `STP` evaluates $f$ two times at each iteration, which means performing two new computations using all the training data for one update of the parameters. In fact, proceeding in such manner is not all the time efficient. In cases when the total number of training samples is extremely large, such as in the case of large scale machine learning, it becomes computationally expensive to use all the dataset at each iteration of the algorithm. Moreover, training an algorithm using minibatches of the data could be as efficient or better than using the full batch as in the case of `SGD` (Gower et al., 2019). Motivated by this, we introduced `MiSTP` to extend `STP` to the case of using subsets of the data at each iteration of the training process.

We consider in this paper the finite-sum problem as it is largely encountered in ML applications, but our approach is applicable to the more general case where we do not have necessarily the finite-sum structure and only an approximation of the objective function can be computed. Such situation may happen, for instance, in the case where the objective function is the output of a stochastic oracle that provides only noisy/stochastic evaluations.

## 1.1 CONTRIBUTIONS

In this section, we highlight the key contributions of this work.

- We propose `MiSTP` method to extend the `STP` method (Bergou et al., 2020) to the case of using only an approximation of the objective function at each iteration.
- We analyse our method's complexity in the case of nonconvex and convex objective function.
- We present experimental results of the performance of `MiSTP` on multiple ML tasks, namely on ridge regression, regularized logistic regression, and training of a neural network. We evaluate the performance of `MiSTP` with different minibatch sizes and in comparison with Stochastic Gradient Descent (`SGD`) (Gower et al., 2019) and other ZO methods.

## 1.2 OUTLINE

The paper is organized as follow: In section 2 we present our `MiSTP` method. In section 2.1 we describe the main assumptions on the random search directions which ensure the convergence of our method. These assumptions are the same as the ones used for `STP` (Bergou et al., 2020). Then, in section 2.2 we formulate the key lemma for the iteration complexity analysis. In section 3 we analyze the worst case complexity of our method for smooth nonconvex and convex problems. In section 4, we present and discuss our experiments results. In section 4.1, we report the results on ridge regression and regularized logistic regression problems, and in section 4.2, we report the result on neural networks. Finally, we conclude in section 5.

### 1.3 NOTATION

Throughout the paper, $\mathcal{D}$ will denote a probability distribution over $\mathbb{R}^d$. We use $\mathbf{E}\left[\cdot\right]$ to denote the expectation, $\mathbf{E}_\xi\left[\cdot\right]$ to denote the expectation over the randomness of $\xi$ conditional to other random quantities, and for two random variables X and Y, $\mathbf{E}[X|Y]$ denotes the expectation of X given Y. $\langle x, y \rangle = x^\top y$ corresponds to the inner product of $x$ and $y$. We denote also by $\|\cdot\|_2$ the $\ell_2$-norm, and by $\|\cdot\|_\mathcal{D}$ a norm dependent on $\mathcal{D}$. We denote by $f_\mathcal{B}$:

$$f_\mathcal{B}(x) = \frac{1}{|\mathcal{B}|} \sum_{i \in \mathcal{B}} f_i(x), \tag{2}$$

where $\mathcal{B}$ is a subset on indexes chosen from the set $[1, 2, \ldots, n]$ and $|\mathcal{B}|$ is its cardinal.

## 2 MiSTP METHOD

Our *minibatch stochastic three points* (`MiSTP`) algorithm is formalized below as Algorithm 1.

---
**Algorithm 1: Minibatch Stochastic Three Points (`MiSTP`)**

---
**Initialization**

        Choose $x_0 \in \mathbb{R}^d$, positive stepsizes $\{\alpha_k\}_{k \geq 0}$, probability distribution $\mathcal{D}$ on $\mathbb{R}^d$.

**For** $k = 0, 1, 2, \ldots$

        1. Generate a random vector $s_k \sim \mathcal{D}$
        2. Choose elements of the subset $\mathcal{B}_k$ u.a.r
        3. Let $x_+ = x_k + \alpha_k s_k$ and $x_- = x_k - \alpha_k s_k$
        4. $x_{k+1} = \arg\min\{f_{\mathcal{B}_k}(x_-), f_{\mathcal{B}_k}(x_+), f_{\mathcal{B}_k}(x_k)\}$

---

Due to the randomness of the search directions $s_k$ and the minibatches $\mathcal{B}_k$ for $k \geq 0$, the iterates are also random vectors for all $k \geq 1$. The starting point $x_0$ is not random (the initial objective function value $f(x_0)$ is deterministic).

**Lemma 1.** *For $x \in \mathbb{R}^d$ such that $x$ is independent from $\mathcal{B}$, i.e., the choice of $x$ does not depend on the choice of $\mathcal{B}$, $f_\mathcal{B}(x)$ is an unbiased estimator of $f(x)$.*

*Proof.* See appendix $A$, section $A.1$. $\qquad\square$

Throughout the paper, we assume that $f_i$, (for $i = 1, \ldots, n$) is differentiable, and has $L_i$-Lipschitz gradient. We assume also that $f$ is bounded from below.

**Assumption 1.** *The objective function $f_i$, (for $i = 1, \ldots, n$) is $L_i$-smooth with $L_i > 0$ and $f$ is bounded from below by $f_* \in \mathbb{R}$. That is, $f_i$ has a Lipschitz continuous gradient with a Lipschitz constant $L_i$:*

$$\|\nabla f_i(x) - \nabla f_i(y)\|_2 \leq L_i \|x - y\|_2, \qquad \forall x, y \in \mathbb{R}^d$$

*and $f(x) \geq f_*$ for all $x \in \mathbb{R}^d$.*

**Assumption 2.** *We assume that the variance of $f_\mathcal{B}(x)$ is bounded for all $x \in \mathbb{R}^d$:*

$$\mathbf{E}_\mathcal{B}[(f(x) - f_\mathcal{B}(x))^2] < \sigma_{|\mathcal{B}|}^2 < \infty$$

This assumption is very common in the stochastic optimization literature (Larson et al., 2019, section 6). Note that we put the subscript $|\mathcal{B}|$ in $\sigma_{|\mathcal{B}|}$ to mention that this deviation may be dependent on the minibatch size. Consider, for example, the case of sampling minibatches uniformly with replacement. In such case, the expected deviation between $f$ and $f_\mathcal{B}$ satisfy $\mathbf{E}_\mathcal{B}[(f(x) - f_\mathcal{B}(x))^2] \leq \frac{A}{|\mathcal{B}|}$ for all $x \in \mathbb{R}^d$ independent from $\mathcal{B}$ where $A = \sup_{x \in \mathbb{R}^d} \frac{1}{n} \sum_{i=1}^n (f_i(x) - f(x))^2$ (See appendix $A$, section $A.2$). Note that, given that the function $f(y) = y^2$ is convex on $\mathbb{R}$ and using Jensen's inequality we have: $(\mathbf{E}_\mathcal{B}[|f(x) - f_\mathcal{B}(x)|])^2 \leq \mathbf{E}_\mathcal{B}[(|f(x) - f_\mathcal{B}(x)|)^2]$. Therefore, $\mathbf{E}_\mathcal{B}[|f(x) - f_\mathcal{B}(x)|] \leq \sigma_{|\mathcal{B}|}$.

## 2.1 ASSUMPTION ON THE DIRECTIONS

Our analysis in the sequel of the paper will be based on the following key assumption.

**Assumption 3.** *The probability distribution $\mathcal{D}$ on $\mathbb{R}^d$ has the following properties:*

1. *The quantity $\mathbf{E}_{s \sim \mathcal{D}} \|s\|_2^2$ is positive and finite. Without loss of generality, in the rest of this paper we assume that it is equal to 1.*

2. *There is a constant $\mu_{\mathcal{D}} > 0$ and norm $\|\cdot\|_{\mathcal{D}}$ on $\mathbb{R}^d$ such that for all $g \in \mathbb{R}^d$,*

$$\mathbf{E}_{s \sim \mathcal{D}} |\langle g, s \rangle| \geq \mu_{\mathcal{D}} \|g\|_{\mathcal{D}}. \tag{3}$$

As proved in the STP paper (Bergou et al., 2020), multiple distributions satisfy this assumption. For example: the uniform distribution on the unit sphere in $\mathbb{R}^d$, the normal distribution with zero mean and $d \times d$ identity as the covariance matrix, the uniform distribution over standard unit basis vectors $\{e_1, ..., e_d\}$ in $\mathbb{R}^d$, the distribution on $S = s_1, ..., s_d$ where $\{s_1, ..., s_d\}$ form an orthonormal basis of $\mathbb{R}^d$.

## 2.2 KEY LEMMA

Now, we establish the key result which will be used to prove the main properties of our algorithm.

**Lemma 2.** *If Assumptions 1, 2, and 3 hold, then for all $k \geq 0$,*

$$\theta_{k+1} \leq \theta_k - \mu_{\mathcal{D}} \alpha_k g_k + \frac{L}{2} \alpha_k^2 + \sigma_{|\mathcal{B}|}, \tag{4}$$

*where $L_{\mathcal{B}_k} = \frac{1}{|\mathcal{B}_k|} \sum_{i \in \mathcal{B}_k} L_i$, $L = \mathbf{E}[L_{\mathcal{B}_k}] = \frac{1}{n} \sum_{i=1}^n L_i$, $\theta_k = \mathbf{E}[f(x_k)]$ and $g_k = \mathbf{E}[\|\nabla f(x_k)\|_{\mathcal{D}}]$, and $|\mathcal{B}_k|$ is the minibatch size .*

*Proof.* We have: $f(x_{k+1}) - f_{\mathcal{B}_k}(x_{k+1}) \leq |f(x_{k+1}) - f_{\mathcal{B}_k}(x_{k+1})|$ i.e., $f(x_{k+1}) \leq f_{\mathcal{B}_k}(x_{k+1}) + |f(x_{k+1}) - f_{\mathcal{B}_k}(x_{k+1})|$ (5)

We have: $x_{k+1} = \arg\min\{f_{\mathcal{B}_k}(x_k - \alpha_k s_k), f_{\mathcal{B}_k}(x_k + \alpha_k s_k), f_{\mathcal{B}_k}(x_k)\}$, therefore: $f_{\mathcal{B}_k}(x_{k+1}) \leq f_{\mathcal{B}_k}(x_k + \alpha_k s_k)$ (6). From $L_i$-smoothness of $f_i$ we have:

$$f_i(x_k + \alpha_k s_k) \leq f_i(x_k) + \langle \nabla f_i(x_k), \alpha_k s_k \rangle + \frac{L_i}{2} \|\alpha_k s_k\|_2^2$$

By summing over $f_i$ for $i \in \mathcal{B}_k$ and multiplying by $1/|\mathcal{B}_k|$ we get:

$$
\begin{aligned}
f_{\mathcal{B}_k}(x_k + \alpha_k s_k) &\leq f_{\mathcal{B}_k}(x_k) + \langle \nabla f_{\mathcal{B}_k}(x_k), \alpha_k s_k \rangle + \frac{L_{\mathcal{B}_k}}{2} \|\alpha_k s_k\|_2^2 \\
&= f_{\mathcal{B}_k}(x_k) + \alpha_k \langle \nabla f_{\mathcal{B}_k}(x_k), s_k \rangle + \frac{L_{\mathcal{B}_k}}{2} \alpha_k^2 \|s_k\|_2^2 \quad (8)
\end{aligned}
$$

By using inequalities (5), (6), and (8) we get:

$$f(x_{k+1}) \leq f_{\mathcal{B}_k}(x_k) + \alpha_k \langle \nabla f_{\mathcal{B}_k}(x_k), s_k \rangle + \frac{L_{\mathcal{B}_k}}{2} \alpha_k^2 \|s_k\|_2^2 + e_{\mathcal{B}_k}^{k+1}$$

where $e_{\mathcal{B}_k}^{k+1} = |f(x_{k+1}) - f_{\mathcal{B}_k}(x_{k+1})|$

By taking the expectation conditioned on $x_k$ and $s_k$ and using assumption 2 we get:

$$\mathbf{E}[f(x_{k+1})|x_k, s_k] \leq f(x_k) + \alpha_k \langle \nabla f(x_k), s_k \rangle + \frac{L}{2} \alpha_k^2 \|s_k\|_2^2 + \sigma_{|\mathcal{B}|}$$

Similarly, we can get (see details in appendix $A$, section $A.3$):

$$\mathbf{E}[f(x_{k+1})|x_k, s_k] \leq f(x_k) - \alpha_k \langle \nabla f(x_k), s_k \rangle + \frac{L}{2} \alpha_k^2 \|s_k\|_2^2 + \sigma_{|\mathcal{B}|}$$

From the two inequalities above we conclude:

$$\mathbf{E}[f(x_{k+1})|x_k, s_k] \leq f(x_k) - \alpha_k |\langle \nabla f(x_k), s_k \rangle| + \frac{L}{2} \alpha_k^2 \|s_k\|_2^2 + \sigma_{|\mathcal{B}|}$$

By taking the expectation over $s_k$ and using inequality (3) we get:

$$\mathbf{E}[f(x_{k+1})|x_k] \leq f(x_k) - \alpha_k \mu_{\mathcal{D}} \|\nabla f(x_k)\|_{\mathcal{D}} + \frac{L}{2}\alpha_k^2 + \sigma_{|\mathcal{B}|}$$

By taking expectation in the above inequality and due to the tower property of the expectation we get:

$$\mathbf{E}[f(x_{k+1})] \leq \mathbf{E}[f(x_k)] - \alpha_k \mu_{\mathcal{D}} \mathbf{E}[\|\nabla f(x_k)\|_{\mathcal{D}}] + \frac{L}{2}\alpha_k^2 + \sigma_{|\mathcal{B}|}$$

$\square$

## 3 COMPLEXITY ANALYSIS

We first state, in theorem 1, the most general complexity result of `MiSTP` where we do not make any additional assumptions on the objective functions besides smoothness of $f_i$, for $i = 1, \ldots, n$, and boundedness of $f$. The proofs follow the same reasoning as the ones in `STP` (Bergou et al., 2020), we defer them to the appendix.

**Theorem 1** (nonconvex case). *Let Assumptions 1, 2, and 3 be satisfied and $\sigma_{|\mathcal{B}|} < \frac{(\mu_{\mathcal{D}}\epsilon)^2}{2L}$. Choose a fixed stepsize $\alpha_k = \alpha$ with $(\mu_{\mathcal{D}}\epsilon - \sqrt{(\mu_{\mathcal{D}}\epsilon)^2 - 2L\sigma_{|\mathcal{B}|}})/L < \alpha < (\mu_{\mathcal{D}}\epsilon + \sqrt{(\mu_{\mathcal{D}}\epsilon)^2 - 2L\sigma_{|\mathcal{B}|}})/L$, If*

$$K \geq k(\varepsilon) \stackrel{def}{=} \left\lceil \frac{f(x_0) - f_*}{\mu_{\mathcal{D}}\varepsilon\alpha - \frac{L}{2}\alpha^2 - \sigma_{|\mathcal{B}|}} \right\rceil - 1, \tag{8}$$

*then $\min_{k=0,1,\ldots,K} \mathbf{E}[\|\nabla f(x_k)\|_{\mathcal{D}}] \leq \varepsilon$. In particular, we have: $\alpha_{optimal} = \mu_{\mathcal{D}}\varepsilon/L$*

*Proof.* see appendix $A$, section $A.4$ $\square$

We now state the complexity of `MiSTP` in the case of convex $f$. To do so, we add the following assumption:

**Assumption 4.** *We assume that $f$ is convex, has a minimizer $x_*$, and has bounded level set at $x_0$:*

$$R_0 \stackrel{def}{=} \max\{\|x - x_*\|_{\mathcal{D}}^* \ : \ f(x) \leq f(x_0)\} < +\infty,$$

*where $\|\xi\|_{\mathcal{D}}^* \stackrel{def}{=} \max\{\langle \xi, x \rangle \mid \|x\|_{\mathcal{D}} \leq 1\}$ defines the dual norm to $\|\cdot\|_{\mathcal{D}}$.*

Note that if the above assumption holds, then whenever $f(x) \leq f(x_0)$, we get $f(x) - f(x_*) \leq \langle \nabla f(x), x - x_* \rangle = \|\nabla f(x)\|_{\mathcal{D}}(x - x_*)^T \nabla f(x)/\|\nabla f(x)\|_{\mathcal{D}} \leq \|\nabla f(x)\|_{\mathcal{D}}\|x - x_*\|_{\mathcal{D}}^* \leq R_0\|\nabla f(x)\|_{\mathcal{D}}$. That is,

$$\|\nabla f(x)\|_{\mathcal{D}} \geq \frac{f(x) - f(x_*)}{R_0}. \tag{9}$$

**Theorem 2** (convex case). *Let Assumptions 1, 2, 3, and 4 be satisfied. Let $\varepsilon > 0$ and $\sigma_{|\mathcal{B}|} < \frac{(\mu_{\mathcal{D}}\epsilon)^2}{4LR_0^2}$, choose constant stepsize $\alpha_k = \alpha = \frac{\varepsilon\mu_{\mathcal{D}}}{LR_0}$, If*

$$K \geq \frac{LR_0^2}{\mu_{\mathcal{D}}^2\varepsilon} \log\left(\frac{4(f(x_0) - f(x_*))}{\varepsilon}\right), \tag{10}$$

*then $\mathbf{E}[f(x_K) - f(x_*)] \leq \varepsilon$.*

*Proof.* see appendix $A$, section $A.5$ $\square$

## 4 NUMERICAL RESULTS

In this section, we report the results of some experiments conducted in order to evaluate the efficiency of `MiSTP`. All the presented results are averaged over 10 runs of the algorithm and the confidence intervals (the shaded region in the graphs) are given by $\mu \pm \frac{\sigma}{2}$ where $\mu$ is the mean and $\sigma$ is the standard deviation. For each minibatch size, we choose the learning rate $\alpha$ by performing

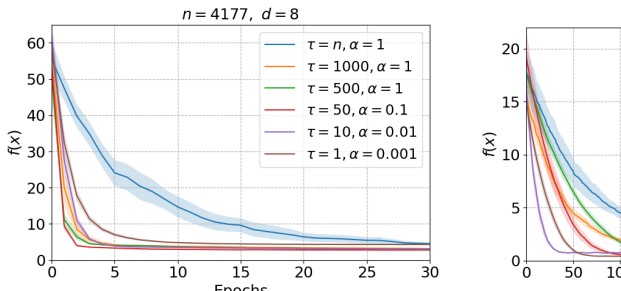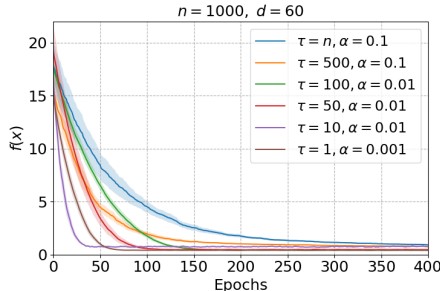

Figure 1: Performance of MiSTP with different minibatch sizes on ridge regression problem. On the left, the abalone dataset. On the right, the splice dataset.

a grid search on the values 1,0.1,0.01,... and select the one that gives the best performance. $\tau$ denotes the minibatch size, i.e., $\tau = |\mathcal{B}|$. In all our implementations, the starting point $x_0$ is sampled from the standard Gaussian distribution. The distribution $\mathcal{D}$ used to sample search directions, unless specified otherwise, is the normal distribution with zero mean and $d \times d$ identity as the covariance matrix.

### 4.1 MiSTP ON RIDGE REGRESSION AND REGULARIZED LOGISTIC REGRESSION PROBLEMS

We performed experiments on ridge regression and regularized logistic regression. They are problems with strongly convex objective function $f$.

In the case of ridge regression we solve:

$$\min_{x \in \mathbb{R}^d} \left[ f(x) = \frac{1}{2n} \sum_{i=1}^{n} (A[i,:]x - y_i)^2 + \frac{\lambda}{2} \|x\|_2^2 \right] \tag{11}$$

and in the case of regularized logistic regression we solve:

$$\min_{x \in \mathbb{R}^d} \left[ f(x) = \frac{1}{2n} \sum_{i=1}^{n} \ln(1 + \exp(-y_i A[i,:]x)) + \frac{\lambda}{2} \|x\|_2^2 \right] \tag{12}$$

In both problems $A \in \mathbb{R}^{n \times d}$, $y \in \mathbb{R}^n$ are the given data and $\lambda > 0$ is the regularization parameter. For logistic regression: $y \in \{-1, 1\}^n$ and all the values in the first column of $A$ are equal to 1. [1] For both problems we set $\lambda = 1/n$. The experiments of this section are conducted using LIBSVM datasets (Chang & Lin, 2011).

In section 4.1.1, we evaluate the performance of MiSTP when using different minibatch sizes. In section 4.1.2 we evaluate the performance of MiSTP compared to SGD, and in section 4.1.3 we compare the performance of MiSTP with some other ZO methods.

### 4.1.1 MiSTP WITH DIFFERENT MINIBATCH SIZES

Figures 1 and 2 show the performance of MiSTP when using different minibatch sizes. From these figures we see good performance of MiSTP. For different minibatch sizes, it generally converges faster than the original STP (the full batch) in terms of number of epochs. We notice also that there is an optimal minibatch size that gives the best performance for each dataset: among the tested values, for the 'abalone' dataset it is equal to 50, for 'splice' dataset it is 1, for 'a1a' and 'australian' datasets it is 10. All those optimal minibatch sizes are just a very small subset of the whole dataset which results in less computation at each iteration. Those results also show that we could get a good performance when using only an approximation of the objective function using a small subset of the data rather than the exact function evaluations.

---

[1] It is known that the value of the first feature must be 1 for all the training inputs when performing logistic regression. When using LIBSVM datasets we add this column to the data.

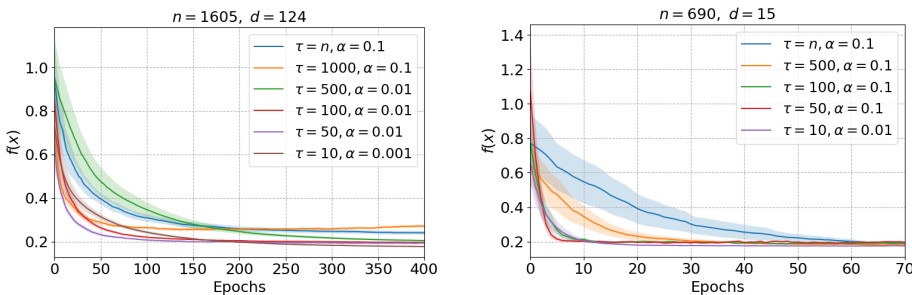

Figure 2: Performance of `MiSTP` with different minibatch sizes on regularized logistic regression problem. On the left, the a1a dataset. On the right, the australian dataset.

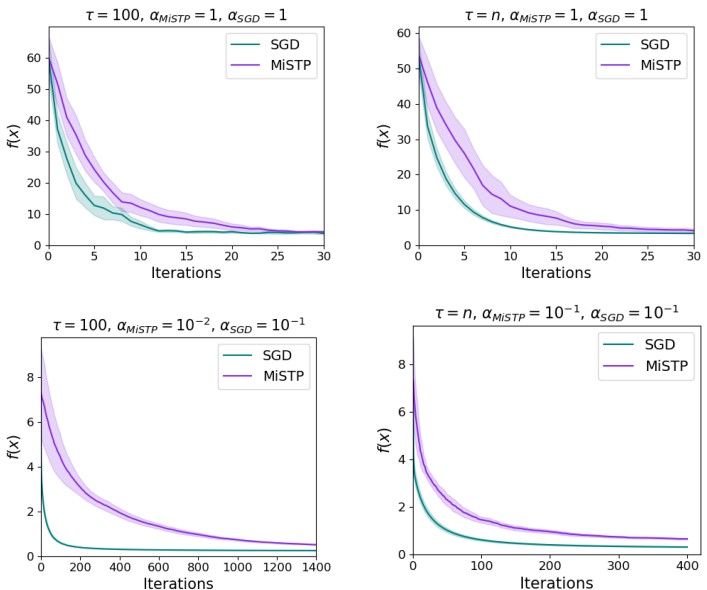

Figure 3: Performance of `MiSTP` and `SGD` on ridge regression problem using real data from LIB-SVM. Above, abalone dataset: $n = 4177$ and $d = 8$. Below, a1a dataset: $n = 1605$ and $d = 123$.

### 4.1.2 MiSTP vs. SGD

In this section we report some results of experiments conducted in order to compare the performance of `MiSTP` to `SGD`. For both methods, we used the same starting point at each run and the same minibatch at each iteration.

Figures 3 and 4 show results of experiments on ridge regression and regularized logistic regression problems respectively. More results are presented in Appendix B. From these experiments we see that in most of the cases, `MiSTP` is able to converge to a good approximation or exactly the same solution as `SGD`. `MiSTP` also gives competitive performance to `SGD` when the dimension of the problem is small, i.e., $d$ is less or around 10. When the dimension of the problem is big, i.e., $d$ is of order of tens, `MiSTP` needs more iterations compared to `SGD` to converge to just an approximation of the solution. In all cases, we see that the number of iterations that `MiSTP` needs to converge increases as the batch size decreases. It also increases as the dimension of the problem increases while `SGD` is slightly affected by this. In Appendix B, we report the values of the approximation $f_{\mathcal{B}}$ alongside $f$ for multiple minibatch sizes. We can see that starting from a given batch size (generally when $\tau \geq 500$ for the given datasets) $f_{\mathcal{B}}$ is a good approximation of $f$ which shows that we can get the same results when training a model with only a subset of the data as when using all available samples. Consequently, this results in less computations.

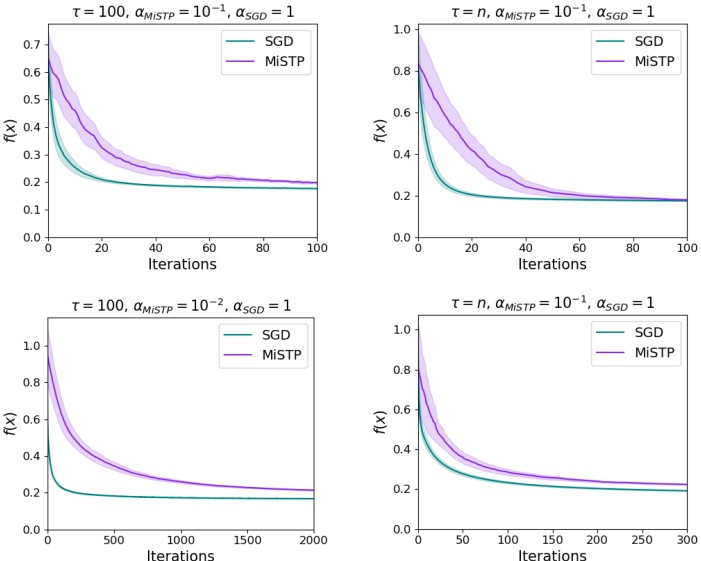

Figure 4: Performance of `MiSTP` and `SGD` on regularized logistic regression problem using real data from LIBSVM. Above, australian dataset : $n = 690$ and $d = 15$. Below, a1a dataset : $n = 1605$ and $d = 124$.

### 4.1.3 MiSTP vs. other zero-order methods

In this section, we compare the performance of `MiSTP` with three other ZO optimization methods. The first is `RSGF`, proposed by Ghadimi & Lan (2013). In this method, at iteration $k$, the iterate is updated as follow:

$$x_{k+1} = x_k - \alpha_k \frac{f_{\mathcal{B}_k}(x_k + \mu_k s_k) - f_{\mathcal{B}_k}(x_k)}{\mu_k} s_k \tag{13}$$

where $\mu_k \in (0,1)$ is the finite differences parameter, $\alpha_k$ is the stepsize, $s_k$ is a random vector following the uniform distribution on the unit sphere, and $\mathcal{B}_k$ is a randomly chosen minibatch. The second is `ZO-SVRG` proposed by Liu et al. (2018, Algorithm 2). For this method, at iteration $k$, the gradient estimation of $f_{\mathcal{B}_k}$ at $x_k$ is given by:

$$\hat{\nabla} f_{\mathcal{B}_k}(x_k) = \frac{d}{\mu}(f_{\mathcal{B}_k}(x_k + \mu s_k) - f_{\mathcal{B}_k}(x_k)) s_k \tag{14}$$

where $\mu > 0$ is the smoothing parameter and $s_k$ is a random direction drawn from the uniform distribution over the unit sphere. And the last is `ZO-CD` (ZO coordinates descent method), in this method, at iteration $k$, the iterate is updated as follow:

$$x_{k+1} = x_k - \alpha_k g_{\mathcal{B}_k}, \qquad g_{\mathcal{B}_k} = \sum_{i=1}^{d} \frac{f_{\mathcal{B}_k}(x_k + \mu e_i) - f_{\mathcal{B}_k}(x_k - \mu e_i)}{2\mu} e_i \tag{15}$$

where $\mu > 0$ is a smoothing parameter and $e_i \in \mathbb{R}^d$ for $i \in [d]$ is a standard basis vector with 1 at its $i$th coordinate and 0 elsewhere.

The distribution $\mathcal{D}$ used here for `MiSTP` is the uniform distribution on the unit sphere. For `RSGF`, `ZO-SVRG`, and `ZO-CD`, we chose $\mu_k = \mu = 10^{-4}$

Figure (5) shows the objective function values against the number of function queries of the different ZO methods using different minibatch sizes. Note that one function query is the evaluation of one $f_i$ for $i \in [n]$ at a given point. From figure (5) we see that, on the ridge regression problem, `MiSTP`, `RSGF`, and `ZO-CD` show competitive performance while `ZO-SVRG` needs much more function queries to converge. On the regularized logistic regression problem, `MiSTP` outperforms all the other methods. `RSGF`, `ZO-CD`, and `ZO-SVRG` need almost 5 times function queries to converge than `MiSTP` for $\tau = 100$ and around 2 times more function queries than `MiSTP` for $\tau = 50$.

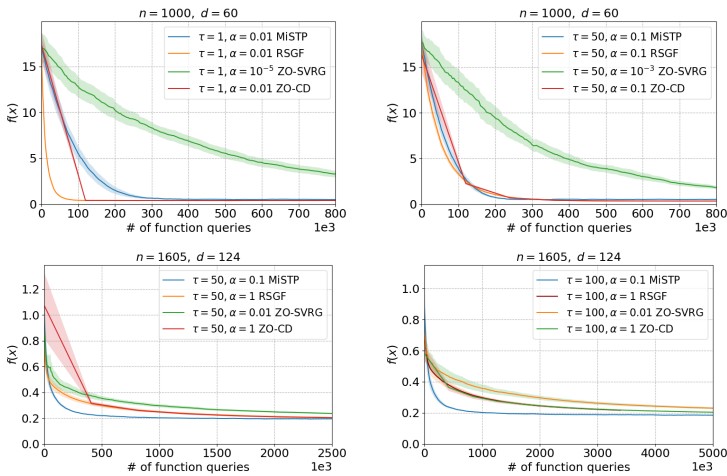

Figure 5: Comparison of `MiSTP`, `RSGF`, `ZO-SVRG`, and `ZO-CD`. Above: ridge regression problem using the splice dataset. Below: regularized logistic regression problem using the a1a dataset.

### 4.2 MiSTP IN NEURAL NETWORKS

Figure 6 shows the results of experiments using `MiSTP` as the optimizer in a multi-layer neural network (NN) for MNIST digit (LeCun et al., 1998) classification with different minibatch sizes. The architecture we used has three fully-connected layers of size 256, 128, 10, with ReLU activation after the first two layers and a Softmax activation function after the last layer. The loss function is the categorical cross entropy. From figure 6 we observe that the minibatch size 6000 outperforms the minibatch size 3000 and the full batch, it converges faster to better accuracy and loss values. $\tau = 6000$ is $1/10$ of the dataset (we used the whole MNIST dataset which has 60000 samples), it leads to less computation time at each iteration than using all the 60000 samples. Besides it largely outperforms the full batch. Those results prove that minibatch training is more efficient than the full batch training and that we can find an optimal minibatch size that leads to efficient training of an NN in terms of performance and computation effort.

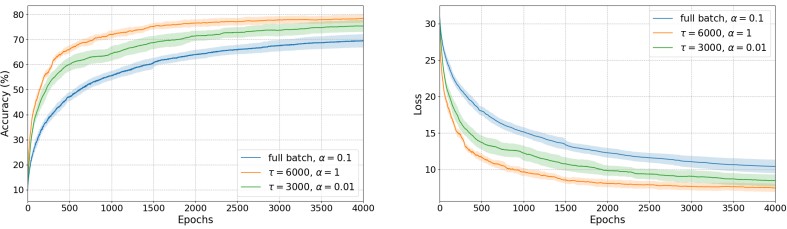

Figure 6: Comparison of different minibatch sizes for `MiSTP` in a multi-layer neural network.

## 5 CONCLUSION

In this paper, we proposed the `MiSTP` method to extend the `STP` method to case of using only an approximation of the objective function at each iteration assuming the error between the objective function and its approximation is bounded. `MiSTP` sample the search directions in the same way as `STP`, but instead of comparing the objective function at three points it compares an approximation. We derived our method's complexity in the case of nonconvex and convex objective function. The presented numerical results showed encouraging performance of `MiSTP`. In some settings, it showed superior performance over the original `STP`. There are a number of interesting future works to further extend our method, namely deriving a rule to find the optimal minibatch size, comparing the performance of `MiSTP` with other zero-order methods on deep neural networks problems, extending `MiSTP` to the case of distributed learning, and investigating `MiSTP` in the non-smooth case.

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
