# OpenReview forum: "Minibatch Stochastic Three Points Method for Unconstrained Smooth Minimization"
_ICLR.cc/2023/Conference — Submitted to ICLR 2023_

### Official Review · Reviewer_KsoD · 2022-10-16

**Confidence:** 4
**Correctness:** 2
**Technical Novelty And Significance:** 2
**Empirical Novelty And Significance:** 2
**Recommendation:** 5

**Clarity, Quality, Novelty And Reproducibility:**

More clarity is needed to understand the motivation, strength and novelty of this work. It lacks empirical results on complicated tasks and comprehensive comparison to other methods in order to make the effectiveness of the proposed algorithm convincing.

**Strength And Weaknesses:**

Strengthes:
Compared to STP, evaluating objective value on a batch of data instead of the full data can make the algorithms much more practical.

Weakness:

1. The motivation of this work is not well justified. What are the advantages of using this STP-type algorithms compared to other existing works such as two-point estimators? Even though STP was proposed in another work, it is still necessary to clearly discuss the motivation in order to help the audience to understand the significance of this work.

2. Given the existing STP work, this submission extends it to a version that only requires an approximation of the objective instead of an accurate evaluation. What are the challenges in the analysis? Can the introduced noise by inaccurate estimation of objective values just be controlled in similar way to in the analysis of stochastic gradient descent? Therefore, I wonder what are the specific challenges and key techniques to solve these challenges in this submission?

3. The experiments in Section 4.1 are for easy problems where gradients can be easily computed and so SGD are applicable. And experiments in Section 4.2 are not compared with other zero order methods. Hence, the true performance of the proposed MiSTP is questionable. Besides, the problems in Section 4.1 are strongly convex which expect better complexity rates than those in Theorem 1 and Theorem 2. Maybe the authors can consider to add the analysis of strongly convex cases as well.




**Summary Of The Paper:**

This submission proposes a minibatch stochastic three points method, which only requires an approximation of the objective function at each iteration. It is an extension of a previous work of stochastic three points method that requires the exact evaluation of objective functions.

**Summary Of The Review:**

I tend to vote for rejection based on the current manuscript being lack of motivation,  not having adequate comparison to existing methods and not showing novelty in terms of analysis techniques.

---

> ### Author Response · Authors · 2022-11-15
> **Response to Reviewer KsoD**
>
> We first thank the reviewer for reviewing our paper and providing comments on our work. We address the comments below.
>
> * Concerning the motivation of our work, STP method [1] is a general variant of direct search methods. Such methods do not compute at all an estimation of the gradient as opposed, for example, to the popular RGF method [2]  that uses the two points gradient estimator.  As proved in [1], STP outperforms the existing direct search methods and is also competitive in performance to RGF method. STP (and also MiSTP) is also different from RGF (and similar methods that estimate the gradient) in that it allows multiple choices of the distribution for sampling the search directions. Besides, STP is very simple, easy to implement, and practical.
>
>   Our goal in this work was to investigate STP in the case of the finite sum problem. We want to make it more efficient in such setting, hence the proposed method MiSTP. We were also motivated by the transition from gradient descent (GD) to stochastic gradient descent (SGD) in the first-order case and the wide impact SGD has and we want to do the same in the zeroth order settings for STP.
>
> * Concerning the challenges in the analysis, as we explained to the second reviewer jzrD, the new iterate in MiSTP depend on two randomnesses: the randomness of the search directions and the randomness in the minibatch selection unlike STP in which we have only the randomness of the search directions. In particular, we have: $E[f_{\mathcal{B}}(x_{k+1})] \neq f(x_{k+1})$ which will not make it possible to apply similar techniques as for STP.
>
> * Concerning the choice of the problems for the experiments, we just selected some classic and popular problems for the purpose of evaluating the behavior of MiSTP and compare its performance with other methods. We also wanted to compare our method to SGD hence the need for problems for which it is possible to compute gradients. We will consider in the future adding also experiments on problems for which it is impossible to compute the gradient.
>
> * Concerning the experiment using a neural network in section 4.2, our goal was to show that MiSTP outperforms the original STP method in such setting. Comparing MiSTP with other similar methods on deep neural networks problems is among our future work as we mentioned in the conclusion.
>
> * Concerning the convergence analysis in the strongly convex case. We investigated this and we found sub-optimal bounds, mainly similar bounds as the convex case. We decided to not include this in the paper.
>
> [1] E. H. Bergou, E. Gorbunov, and P. Richtárik. Stochastic three points method for unconstrained
> smooth minimization. SIAM Journal on Optimization, 30(4):2726–2749, 2020.
>
> [2] Y. Nesterov and V. Spokoiny. Random gradient-free minimization of convex functions. Foundations of Computational Mathematics, 17:527–566, 2017.

---

### Official Review · Reviewer_o1C6 · 2022-10-24

**Confidence:** 5
**Correctness:** 3
**Technical Novelty And Significance:** 1
**Empirical Novelty And Significance:** 1
**Recommendation:** 3

**Clarity, Quality, Novelty And Reproducibility:**

The idea has been used before for the deterministic problems and the setting considered in this submission is not truly stochastic.

**Strength And Weaknesses:**

The idea of the manuscript is similar to that of (Bergou et al., 2020) and the only difference is that the approximation of the objective function is considered for the three-point estimate over mini-batches. However, the authors assume that they can calculate the objective function at different points over a fixed choice of functions from the finite-sum which is not truly stochastic setting. Moreover, no noise is involved in estimating each $f_i$. The sample complexities have been proposed based on $\mu_D$ which hides the dependence on the problem dimension. The numerical experiments have been also conducted just over low-dimensional settings. No comparison with existing results has been presented.

**Summary Of The Paper:**

In this submission, the authors study zeroth-order optimization methods for minimizing deterministic finite-sum problem. In particular, the authors present mini-batch three points zeroth-order method in which the objective function is calculated at three different points at each iteration over a fixed mini-batch of functions from the finite-sum. They also present sample complexity of their proposed method for both convex and nonconvex settings and conduct some numerical experiments to show performance of the algorithm.

**Summary Of The Review:**

The contribution is incremental and the results have not been presented clearly.

---

> ### Author Response · Authors · 2022-11-15
> **Response to Reviewer o1C6**
>
> We first thank the reviewer for reviewing our paper and providing comments on our work. We address the comments below.
>
> * Concerning the stochastic setting, in our paper, we are solving the problem of minimizing a function $f$ which can be defined by  $f=E[f_{\mathcal{B}}]$ which means that we are, in fact, solving a stochastic optimization problem. You can also refer, for example, to the work of [1]. The authors also consider the finite sum structure and make use of only a selected minibatch of $f_i$ to estimate the gradient. The authors also consider that such setting is stochastic. Our setting is also similar to the case of the stochastic gradient descent algorithm. Our approach is also applicable to the more general stochastic case in which we have only the objective function as the output of a stochastic oracle that provides stochastic evaluations. As we mentioned in our introduction, we focused on the finite sum problem as it is largely encountered in machine learning problems.
>
> * Concerning giving the complexities using $\mu_{\mathcal{D}}$ ,  $\mu_{\mathcal{D}}$  is related to the distribution $\mathcal{D}$ used to sample the search directions. Whenever we change the distribution $\mathcal{D}$ $\mu_{\mathcal{D}}$  changes, so it is better to give the complexity using $\mu_{\mathcal{D}}$  which is general. See Lemma 3.4 in [2] for some examples of the values of $\mu_D$.
>
> * Concerning the dimensional settings for the experiments.     Note that ZO/DFO methods usually suffer from the curse of dimension, so they are more adapted to low to medium dimension. However, in our case besides experiments on low dimensional problems (regression problems) we also conducted an experiment on a high dimensional problem ( the classification problem using a fully connected neural network) see section 4.2.
>
> * Concerning the comparison with existing results, we presented some empirical results for the comparison of our method with some other existing zeroth-order methods in section 4.1.3. More exhaustive numerical experiments are left to future work.
>
> [1] S. Liu, B. Kailkhura, P.-Y. Chen, P. Ting, S. Chang, and L. Amini. Zeroth-order stochastic variance reduction for nonconvex optimization. Advances in Neural Information Processing Systems (NeurIPS), pp. 3731–3741, 2018.
>
> [2] E. H. Bergou, E. Gorbunov, and P. Richtárik. Stochastic three points method for unconstrained smooth minimization. SIAM Journal on Optimization, 30(4):2726–2749, 2020

---

### Official Review · Reviewer_jzrD · 2022-10-24

**Confidence:** 3
**Correctness:** 3
**Technical Novelty And Significance:** 3
**Empirical Novelty And Significance:** 2
**Recommendation:** 5

**Clarity, Quality, Novelty And Reproducibility:**

The writing is mostly clear. The proposed method is novel, but the quality of the empirical comparison is below the bar of ICLR.


**Strength And Weaknesses:**

Strength:

1. The paper is well-written and easy to follow.
2. The MiSTP method is simple and easy to implement. It only needs an approximation of the function evaluation.

Weaknesses:

My main concern is about the experiments. All models used in the experiments (ridge regression, regularized logistic regression, and fully connected NN) can be efficiently solved by gradient-based methods. None of these methods require the use of zeroth-order optimization. I suggest the authors perform experiments on some ML applications where computing gradients is intractable. The authors may refer to the experiments of the ZO-SVRG paper(Liu et al., 2018).

other concerns:

1. Thm 1 and 2 require $\sigma_{|B|}<O(\epsilon^2)$, which means that the batch size may be proportional to $\epsilon^{-2}$. Since $\epsilon$ could be very small, the batch size may be very large.
2. I suggest the authors briefly introduce the main difficulty of extending STP to MiSTP.

**Summary Of The Paper:**

This paper studies smooth zeroth-order optimization and extends the STP method to the finite-sum setting. The proposed method is very simple and easy to implement. The authors also provide convergence analysis and empirically verified the efficiency of the proposed method.

**Summary Of The Review:**

Overall, I think the proposed method is interesting, but the experiments part greatly reduces the quality of this paper.

---

After rebuttal:

I'm disappointed that the paper does not contain experiments on problems for which gradients are not available. Thus I will keep my score.

---

> ### Author Response · Authors · 2022-11-15
> **Response to Reviewer jzrD**
>
> We first thank the reviewer for reviewing our paper and providing comments on our work. We address the comments below.
>
> * Concerning the choice of the problems for the experiments, we just selected some classic and popular problems for the purpose of evaluating the behavior of MiSTP and comparing its performance with other methods. We also wanted to compare our method to SGD hence the need for problems for which it is possible to compute gradients. However, we do agree with the reviewer that it would be interesting to also test the method on problems for which gradients are not available. We will consider that in our future work related to MiSTP.
>
> * Concerning the condition on $\sigma_{|\mathcal{B}|}$ in the theorems, we agree with the reviewer that it can mean that the batch size may be very large. We note that MiSTP is using a kind of " biased gradient estimator", so the need of such assumption on $\sigma_{|\mathcal{B}|}$ is to be able to prove the convergence of the algorithm. This is typical in several works in the literature dealing with biased stochastic gradients. See for instance references [1,2,3].
>
> * Concerning the main difficulty in extending STP to MiSTP, in STP, the new iterate depends only on one randomness which is the randomness of the search directions while in MiSTP, besides the randomness of the search direction, the new iterate depend also on the randomness of the minibatch. In particular, we have $E[f_{\mathcal{B}}(x_{k+1})] \neq f(x_{k+1})$ which will not make it possible to apply similar techniques as for STP.
>
> [1] A. S. Bandeira, K. Scheinberg, and L. N. Vicente. Convergence of trust-region methods based on probabilistic models. SIAM Journal on Optimization, 24(3):1238–1264, 2014.
>
> [2] E. Bergou, Y. Diouane, V. Kungurtsev, and C. W. Royer. A stochastic Levenberg-Marquardt method using random models with application to data assimilation. arXiv:1807.02176, 2018.
>
> [3] J. Blanchet, C. Cartis, M. Menickelly, and K. Scheinberg. Convergence Rate Analysis of a Stochastic Trust Region Method for Nonconvex Optimization. arXiv:1609.07428, 2016.

---

### Official Review · Reviewer_dcdP · 2022-10-27

**Confidence:** 3
**Correctness:** 4
**Technical Novelty And Significance:** 2
**Empirical Novelty And Significance:** 2
**Recommendation:** 5

**Clarity, Quality, Novelty And Reproducibility:**


Novelty:

- The paper extended a recently proposed STP algorithm for zeroth order optimization to the finite sum setting. However, the technical extension to stochastic setting seems to be very standard. For example, the Key lemma follows almost the same development of Lemma 3.5 of [1].



[1] Bergou, E.H., Gorbunov, E. and Richtárik, P., 2020. Stochastic three points method for unconstrained smooth minimization. *SIAM Journal on Optimization*, *30*(4), pp.2726-2749.




**Strength And Weaknesses:**

### Strength

1. It's easy to follow and understand the technical details. The paper is well organized and quite readable. The theoretical analysis is sound.

2. The experiments are quite detailed, which clearly demonstrates the great performance of  MiSTP in a wide range of problems.


### Weakness

- Since the paper considers finite sum setting, developing a variance reduced algorithm can substantially enrich the contribution and solidness of the paper

- It is not clear to me what theoretical advantage MiSTP has, compared with existing algorithm such as RSGF or ZO-SVRG.

- In the theoretical part the authors show the complexity under convex and non convex case but the experiment part is about strong convex problem. Can you also develop the rate for strongly convex setting?

A minor issue:
Should we call the algorithm a zeroth-order method as opposed to zero-order method?

**Summary Of The Paper:**

This paper presented the mini-batch extension of STP method in the finite-sum problem. The paper analyzed the complexity of minibatch STP in both non-convex and convex cases and examine the performance of MiSTP under various settings. The experiment showed that MiSTP converged faster than STP in terms of number of epochs. It also compared MiSTP with SGD and showed that MiSTP could converge to a good approximation or exactly the same solution as SGD. Moreover, MiSTP  compares favorably against several state-of-the-art ZO methods.





**Summary Of The Review:**

Overall, the paper provides a novel zeroth-order algorithm for stochastic convex and nonconvex optimization. Both theoretical analysis and experimental results are  well-supported. However, I still have some concern about the technical contribution. I am not fully convinced why the proposed method is more advantageous over existing algorithms. Maybe I missed something, any theoretical or application insights are welcome.

---

> ### Author Response · Authors · 2022-11-15
> **Response to Reviewer  dcdP**
>
> We first thank the reviewer for reviewing our paper and providing comments on our work. We address the comments below.
>
> * Concerning the comparison with existing works such as  RSGF or ZO-SVRG and advantages of MiSTP over them, on one hand, MiSTP has the same convergence rate as RSGF which is $O(\frac{\sqrt{d}}{\sqrt{T}})$ where $T$ is the number of iterations. ZO-SVRG has a convergence rate of $O(\frac{d}{T}+\frac{1}{|\mathcal{B}|})$, where $|\mathcal{B}|$ is the minibatch size, which improves the rate of MiSTP and  RSGF from $O(\frac{\sqrt{d}}{\sqrt{T}})$ to $O(\frac{d}{T})$ but it suffers from an error term of order $O(\frac{1}{|\mathcal{B}|})$. On the other hand,  MiSTP and  RSGF have a query complexity of $O(|\mathcal{B}|T)$ which is very low compared the the query complexity of ZO-SVRG which is $O(nE+|\mathcal{B}|T)$ where $E$ is the number of epochs.
>
>
>     One of the advantages of MiSTP over existing methods such as RSGF or ZO-SVRG is that it allows multiple choices for the distribution used to sample the search direction and not only the uniform distribution over the unit sphere or the Gaussian distribution.  Also, in our numerical results in section 4.1.3, MiSTP is competitive, and in some cases, outperforms the other methods in terms of number of function evaluations. Besides, MiSTP is very simple and easy to use.
>
> * Concerning adding a variance reduction technique to our method, we do agree with reviewer that it can make our method more robust. The first stage of evaluating MiSTP has given the results presented in this paper. We are working on investigating more this method and make it more efficient. Adding a variance reduction technique is among our future work.
>
> * Concerning the convergence analysis in the strongly convex case. We investigated this and we found sub-optimal bounds, mainly similar bounds as the convex case. We decided to not include this in the paper.
>
> * zeroth-order or zero order are interchangeably used to refer to methods that do not compute the derivatives of the objective function.

---

### Decision · Program_Chairs · 2023-01-20

**Decision:**

Reject

**Justification For Why Not Higher Score:**

All models used in the experiments (ridge regression, regularized logistic regression, and fully connected NN) can be efficiently solved by gradient-based methods. None of these methods require the use of zeroth-order optimization: the authors should perform experiments on some ML applications where computing gradients is intractable (see for instance Liu et al., 2018).
The theoretical advantage MiSTP with existing algorithm such as RSGF or ZO-SVRG could be more carefully described.
The differences with Bergou et al. 2020 should be highlighted to underline the current contribution of this work.
Developing a variance reduced algorithm can substantially enrich the contribution of the paper.


**Justification For Why Not Lower Score:**

The writing is mostly clear. The proposed method is novel.

**Metareview: Summary, Strengths And Weaknesses:**

The authors study zeroth-order optimization methods for minimizing deterministic finite-sum problem.
They present mini-batch three points zeroth-order method in which the objective function is calculated at (two) different points at each iteration over a fixed mini-batch of functions from the finite-sum.
Sample complexity for convex and nonconvex settings is proposed and some numerical experiments illustrate the performance of the algorithm.